# Transcriptome analysis of peripheral blood of *Schistosoma mansoni* infected children from the Albert Nile region in Uganda reveals genes implicated in fibrosis pathology

Joyce Namulondo[1], Oscar Asanya Nyangiri[1], Magambo Phillip Kimuda[1], Peter Nambala[2], Jacent Nassuuna[3], Moses Egesa[3,4], Barbara Nerima[2], Savino Biryomumaisho[1], Claire Mack Mugasa[1], Immaculate Nabukenya[1], Drago Kato[1], Alison Elliott[3,4], Harry Noyes[5], Robert Tweyongyere[1], Enock Matovu[1], Julius Mulindwa[2]*, for the TrypanoGEN+ research group of the H3Africa consortium

1 College of Veterinary Medicine, Animal Resources and Biosecurity, Makerere University, Kampala, Uganda, 2 College of Natural Sciences, Makerere University, Kampala, Uganda, 3 Vaccine Research Theme, MRC/UVRI and LSHTM Uganda Research Unit, Entebbe, Uganda, 4 Department of Infection Biology, London School of Hygiene and Tropical Medicine, London, United Kingdom, 5 Centre for Genomic Research, University of Liverpool, United Kingdom

* julius.mulindwa@mak.ac.ug

**Data Availability Statement:** The phenotype and RNA sequence data have been deposited in the

## Abstract

Over 290 million people are infected by schistosomes worldwide. Schistosomiasis control efforts focus on mass drug treatment with praziquantel (PZQ), a drug that kills the adult worm of all *Schistosoma* species. Nonetheless, re-infections have continued to be detected in endemic areas with individuals living in the same area presenting with varying infection intensities. Our objective was to characterize the transcriptome profiles in peripheral blood of children between 10–15 years with varying intensities of *Schistosoma mansoni* infection living along the Albert Nile in Uganda. RNA extracted from peripheral blood collected from 44 *S. mansoni* infected (34 high and 10 low by circulating anodic antigen [CAA] level) and 20 uninfected children was sequenced using Illumina NovaSeq S4 and the reads aligned to the GRCh38 human genome. Differential gene expression analysis was done using DESeq2. Principal component analysis revealed clustering of gene expression by gender when *S. mansoni* infected children were compared with uninfected children. In addition, we identified 14 DEGs between *S. mansoni* infected and uninfected individuals, 56 DEGs between children with high infection intensity and uninfected individuals, 33 DEGs between those with high infection intensity and low infection intensity and no DEGs between those with low infection and uninfected individuals. We also observed upregulation and downregulation of some DEGs that are associated with fibrosis and its regulation. These data suggest expression of fibrosis associated genes as well as genes that regulate fibrosis in *S. mansoni* infection. The relatively few significant DEGS observed in children with schistosomiasis suggests that chronic *S. mansoni* infection is a stealth infection that does not stimulate a strong immune response.

European Genome-Phenome Archive with accession number EGAS00001007173, and can be accessed via the link https://ega-archive.org/search/EGAS00001007173.

**Funding:** This work was supported by Human Heredity and Health in Africa (H3Africa) through Science for Africa foundation (H3A-18-004 to EM). The funders had no role in study design, data collection and analysis, decision to publish, or preparation of the manuscript.

**Competing interests:** The authors have declared that no competing interests exist.

## Author summary

Schistosomiasis is a neglected tropical disease transmitted via an intermediate snail host through contact with contaminated fresh water. Even with routine Mass Drug Administration for treatment of the infection, re-infections are still common and variations in infection intensity and pathology are still observed in individuals in the same location. These may be due to differences in individuals' response to *S. mansoni* infection. In this study, we used RNAseq to identify differentially expressed genes associated with *S. mansoni* infection in children between 10–15 years. We conducted comparisons between phenotypes including infection intensities measured by circulating anodic antigen, wasting by body mass index and stunting by height-for-age Z-score. Our data showed very low numbers of significantly differentially expressed genes in all comparisons. Some of the few differentially expressed genes that were observed were associated with fibrosis which is the cause of pathology in humans and has been observed in late stages of *S. mansoni* infection in murine studies.

## Introduction

Over 290 million people are infected by schistosomes worldwide. The infection is widespread in tropical and sub-tropical regions with over 78 countries reporting transmission [1]. The WHO estimated that approximately 236.6 million people required treatment in 2019 [2]. During their lifespan, schistosomes live in blood vessels and females lay eggs which migrate through and lodge in organs of the host provoking local and systemic responses [1]. Despite mass administration of praziquantel (PZQ) to control the infection, communities still have high prevalence of schistosomiasis with variations in intensity of infection among individuals [3,4]. Schistosomiasis in Uganda has been shown to be mainly caused by *Schistosoma mansoni* [3,5] with low pockets of *Schistosoma haematobium* [6,7]. The WHO recommended choice for schistosomiasis diagnosis is microscopy to detect *S. mansion* eggs in faeces by Kato Katz (KK) and *S. haematobium* eggs in urine by filtration. Microscopy is low cost but labour intensive and has low sensitivity particularly in low endemic areas. Other more sensitive techniques such as the Point of Care–Circulating Cathodic Antigen (POC-CCA) and up-converting phosphor lateral flow- circulating Anodic Antigen (UCP-LF CAA) have been developed for diagnosis of schistosomiasis [8]. POC-CCA is available in a cassette format to detect the Cathodic Circulating Antigen (CCA) secreted by schistosomes in urine within 20 minutes [9]. It is a qualitative test and has been recommended to be used to screen for *S. mansion* infections in the field. The UCP-LF CAA on the other hand detects the Circulating Anodic Antigen (CAA) in blood and urine. This test detects even the lowest schistosomiasis infection [10] but cannot be conducted in the field as it requires processing in a laboratory setting.

We recently reported a high prevalence of *S. mansion* infection and stunting among school age children along the Albert Nile in Uganda, more in boys than in girls, coupled with variation in infection intensity [4]. However, there is limited information on gene expression in humans infected with schistosomes that could underpin the observed varying infection intensity. Variations in infection intensity may be linked to the ability of the host to respond to the infection which may be driven by a number of underlying molecular mechanisms. Animal studies have shown upregulation of immune genes early in the infection and of metabolic related genes later in the infection [11–13] as well as differences in expression profiles between susceptible and less susceptible hosts [14]. Additionally, responses of human males and females

to infection appear to differ with each having distinct sets of DEGs as observed in *Schistosoma haematobium* infection [15]. To date, there is no study that has profiled gene expression in the blood of humans infected with *S. mansoni*. This study set out to characterise the transcriptome profiles of genes expressed in children 10–15 years of age infected with *S. mansoni* along the Albert Nile. We hypothesized that the expression of genes in peripheral blood of *S. mansoni* infected children varies between children with high infection compared to those with low infection intensity and that both would differ from the uninfected.

## Methods

### Ethics statement

The study protocol was reviewed by the institutional review board of the Ministry of Health, Vector Control Division Research and Ethics Committee (Reference No. VCDREC106) and approved by the Uganda National Council for Science and Technology (Reference No. UNCST HS 118). The study was conducted with guidance from the district health officials, including the selection and training of the village health teams that were involved in the mobilisation and recruitment of the children into the study. The objectives, potential risks and benefits of the study were explained to the parents/ guardians who signed informed consent, and later explained to the school age children in English and Alur dialect who provided assent for participation in the study. Written formal consent from parents and written assent from the children were obtained. If a child was observed to have *S. mansoni* eggs in their stool, they were offered free treatment, which consisted of praziquantel at a dosage of 40mg/kg administered by trained Ministry of Health personnel, assisted by a district health worker. POC-CCA results were not used as an indication for treatment.

### Study design and study sites

This was a cross-sectional study carried out in communities along the Albert Nile. Samples were collected from school children aged between 10–15 years in Pakwach District located in the Northern part of Uganda near the Albert Nile. The selected areas of sampling were in sub-counties of Pakwach, Panyingoro, Panyimur and Alwi all of which are within 10km of the Albert Nile.

### Screening and recruitment

The gene expression study was part of a cross sectional study conducted between October and November 2020 to determine schistosomiasis prevalence among primary school children aged between 10 to 15 years. Screening and recruitment were done as previously described [4]. Participants aged between 10–15 years were mobilized and educated about schistosomiasis by village health teams. They were then registered and screened based on ability to provide urine for screening by the point-of-care circulating cathodic antigen (POC-CCA) (Rapid Medical Diagnostics, Pretoria, South Africa, batch No. 191031120). Briefly, 2 drops (100μl) of urine were placed on the POC-CCA test cassette and left at room temperature for 20 minutes prior to visualization. The results were scored by modifying the G scores as previously described [4]. The modified scores included: 0 (G1), trace (G2, G3), 1+ (G4, G5), 2+ (G6, G7), 3+ (G8, G9) or 4+ (G10). Selection for inclusion in the RNAseq study was purposively done based on the POC-CCA scores. These included individuals with high POC-CCA (4+ and 3+), low (1+ and trace) and negative (0). Samples were also tested for *S. mansoni* and other parasites eggs in stool using Kato Katz (KK) (**S1 Table**) as earlier published [4]. Samples were later classified by infection intensity using CAA in the laboratory as described below. Participants who provided

urine for screening were recruited to the study after being interviewed and informed consent signed by the parent/guardian and assent by the child. The height and weight were obtained from each participant to obtain estimates of body mass index (BMI) and stunting (Height for Age Z-scores HAZ) as previously described [4].

## Sample collection

POC-CCA was used in the field to classify participants by infection intensity to select and collect blood samples that were sequenced for RNAseq. Following the interview, each selected participant was requested to provide peripheral blood which was collected in PAXgene® Blood RNA (PreAnalytiX, US) tubes for transcriptional analysis and in EDTA tubes (BD Biosciences, US) for plasma separation for circulating anodic antigen (CAA) analysis and for DNA extraction.

## CAA assay for *S. mansoni* infection

To classify infection intensity, Circulating Anodic Antigen (CAA) levels were measured in plasma using the up-converting phosphor lateral flow- circulating Anodic Antigen (UCP-LF CAA) SCAA 20 assay as described previously by Mulindwa and colleagues [4]. Briefly, 50 µl of plasma was mixed with 50 µl of 4% trichloroacetic acid (TCA), vortexed and incubated for 5 minutes at room temperature. Following centrifugation at 13000rpm for 5 minutes, 20µl of the supernatant was incubated with 100 µl CAA reporter conjugate for 1 hour at 37˚C. This was followed by incubation of CAA lateral flow strips. The strips were left to dry and quantified using the Labrox Upcon scanner (Labrox Oy, Finland) from which CAA concentrations were calculated in pg/ml. CAA concentrations > 30 pg/mL were classified as positive; negative (CAA < 30 pg/mL), low infection intensity (CAA 30–1000 pg/mL) and high infection intensity (CAA > 1000 pg/mL) adapted from Corstjens et al [16] as well as guided by email communication from the Leiden assay development team.

## RNA extraction and purification

RNA was extracted from blood collected in PAXgene® Blood RNA tubes using Trizol (Invitrogen, USA) protocol [17,18]. The RNA was quantified using Qubit (Invitrogen, USA) and samples with >1µg of RNA were shipped to the Centre for Genomics Research at the University of Liverpool for sequencing where the quality of the samples was checked using an Agilent Bioanalyser.

## Library preparation and RNASeq

The QIAseq FastSelect rRNA HMR kits (Qiagen) were used to remove rRNA from total RNA and libraries prepared using the NEBNext Ultra II Directional RNA Library Prep Kit (NEB, New England Biolabs). The libraries were sequenced on an Illumina NovaSeq S4 (Illumina) in the 2x150bp read configuration to a target depth of 30 million read pairs per sample at the Centre for Genomic Research at the University of Liverpool. FASTQ reads were aligned to the GRCh38 release 84 human genome sequence obtained from Ensembl [19] using HiSat2 [20] and annotated using the human genome reference, *Homo_sapiens* GRCh38.104 from Ensembl.

## Identification of DEGs

Differentially expressed genes between phenotypes were identified using DESeq2 [21]. Analysis of read counts for each gene considered CAA as the independent variable with age and sex

as covariates. Principal component analysis was done using PCA Explorer to identify samples that appeared as outliers [22]. Genes with adjusted p-value <0.05, Log2 (FC)> 1.0 for up-regulated genes and Log2 (FC) < -0.8 for down regulated genes were selected as significant differentially expressed. We compared gene expression between different infection intensities by pairwise analysis of the different infection intensities described above; 1) all infected vs uninfected (IU), 2) high infection intensity vs low infection intensity (HL), 3) high infection intensity vs uninfected (HU) and 4) low infection intensity vs uninfected (LU) while including sex and age as covariates. As the objective of the study was hypothesis generating, no additional statistical correction was made to the adjusted p-values from DESeq2 for the four comparisons that were analysed.

In addition to high prevalence of schistosomiasis, our previous study found high levels of stunting and under nutrition in this study population [4]. To understand the association of BMI and stunting with gene expression, we conducted linear regression analysis with BMI and stunting as dependent variables and sex and age as covariates for each analysis.

To identify the changes in relative abundance of different cell types with *S. mansoni* infection, the proportions of different cell types in each sample were estimated from the expression data using Bisque [23]. Single cell reference sequence data from bone marrow and peripheral blood from Chinese donors was obtained from 7551 individual human blood cells representing 32 leukocyte cell types [24].

## Results

Following the screening of 914 children aged between 10–15 years in Pakwach District, 727 children were recruited for further studies [4] of which 152 children were recruited to participate in this gene expression by RNAseq study. The samples were selected for RNAseq based on CCA. *Schistosoma mansoni* eggs were found in the samples for RNAseq, however no other parasite eggs were detected by KK in these samples (**S1 Table**). The 152 children with extreme CCA values had blood collected in PAXgene tubes of which eighty-one (81) had the required amount of RNA for sequencing recommended by the sequencing laboratory. One (1) sample did not pass QC and was not sequenced. Eighty (80) samples were therefore sequenced to represent the extremes of infection intensity. Of the 80 sequenced, 11 lacked CAA results and were excluded (**Fig 1**). PCA analysis identified five samples (3 high infection intensity, 2 negative) which did not cluster closely with the remaining samples (**S1 Fig**). As we do not know why these samples did not cluster with the others, we took a conservative approach and excluded them from the analysis. Therefore, 64 samples were analysed for gene expression in peripheral blood of children aged between 10–15 years of which; 34 (17: high infection intensity, 5: low infection intensity, 12: uninfected) were male and 30 (17: high infection intensity, 5: low infection intensity, 8: uninfected) were female (**Tables 1 and S1).**

### Differential expression of genes by comparing *S. mansoni* infection intensity categories

PCA analysis of gene expression data showed clear separation by gender on principal components 1 and 2 (**Fig 2**). Genes were defined as significantly differentially expressed between conditions using the following criteria: adjusted p-value <0.05; fold change (FC) Log2 (FC) > 1.0 for up-regulated genes and Log2 (FC) < -0.8 for down regulated genes. Four contrasts: 1) all infected vs uninfected (IU), 2) high infection vs uninfected (HU), 3) high infection vs low infection (HL), and 4) low infection vs uninfected (LU) were examined. The numbers of differentially expressed up and down regulated genes for each contrast are shown in **Table 2**.

 *Schistosoma mansoni* infected children reveal genes implicated in fibrosis

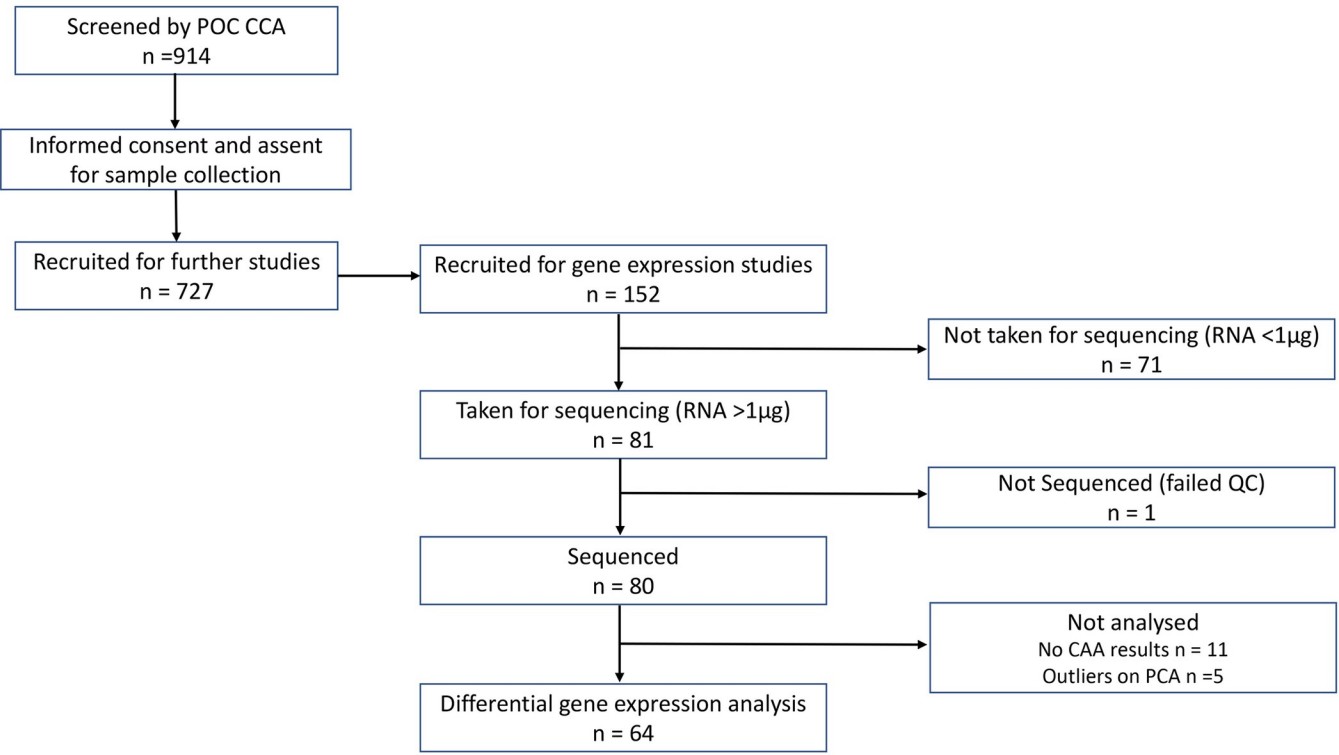

**Fig 1. Screening and recruitment of children for differential gene expression.** Of 727 children recruited for the schistosomiasis study project following consent and assent, 152 were recruited for gene expression studies. Seventy-one (71) were excluded for not having the required amount of RNA required for sequencing hence 81 samples from recruited children were sequenced. One (1) sample did not pass the QC; therefore 80 samples were sequenced; 64 samples were analysed for differentially expressed genes following removal of 5 outliers and 11 samples without CAA results.

## Gene expression differences between the *S. mansoni* infection comparisons

We found that 14 genes were significant differentially expressed between infected and uninfected children of which 9 (64%) were upregulated and 5 (36%) were down regulated among infected individuals compared to uninfected (IU) (**Tables 2 and 3** and **Fig 3A).**

We identified 56 significant DEGs (**Table 2**) listed in **Table 4** among the children with high *S. mansoni* infection compared to the uninfected of which 43 (77%) were upregulated and 13 (23%) were downregulated (**Fig 3B**). We identified 33 significant DEGs (**S2 Table** and **Fig 3C**) among individuals with high infection compared with those with low infection (HL) of which 30 (90.9%) were upregulated and 3 (9.1%) downregulated. We observed DEGs that were common to multiple comparisons and some that were unique to only one comparison (**Fig 4**). One (1) DEG, CCDC168 was upregulated in the three comparisons (IU, HU and HL). Additionally,

**Table 1. Summary of the study sample data by sex and *S. mansoni* infection status using CAA concentration in pg/ml.** There was no significant differences in the number of participants in each infection group by sex (Chisq, p = 0.49) or age (ANOVA, p = 0.18).

| Sex Infection Intensity | Male (n = 34) | Female (n = 30) | Total (n = 64) |
|---|---|---|---|
| **High (>1000 pg/ml)** | 17 | 17 | 34 |
| **Low (25 to <1000 pg/ml)** | 5 | 5 | 10 |
| **Negative (<25 pg/ml)** | 12 | 8 | 20 |
| **Total** | **34** | **30** | **64** |

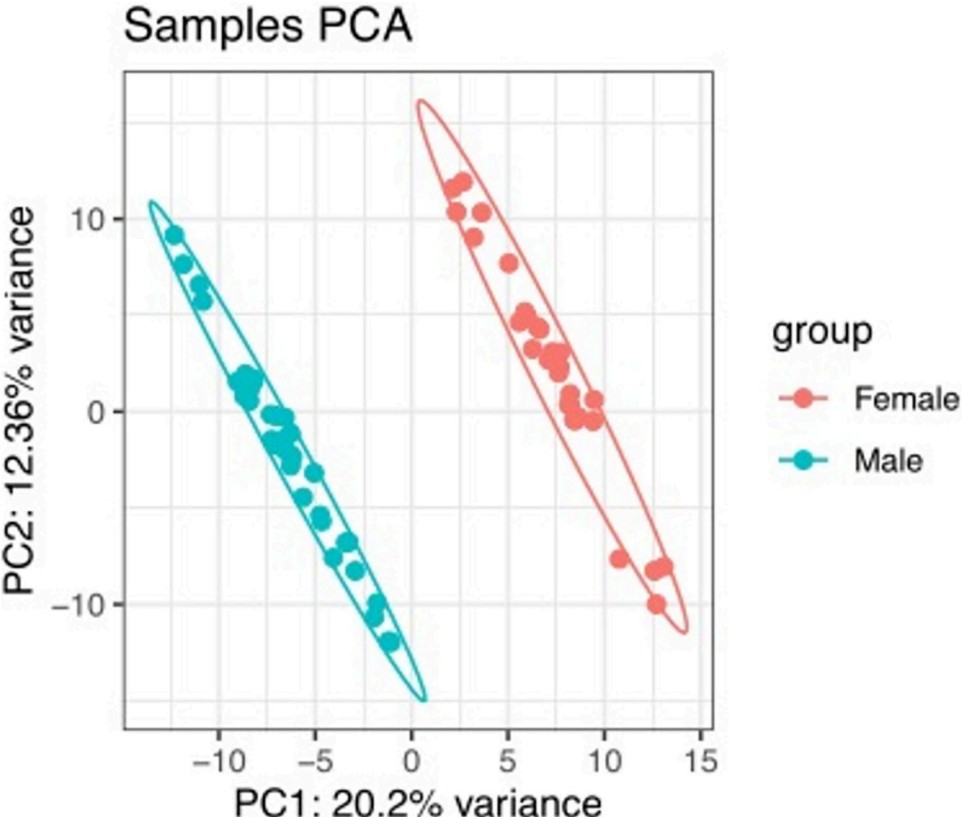

**Fig 2. PCA showed clustering of differentially expressed genes by gender when *S. mansoni* infected children were compared to uninfected.**

7 DEGs, 5 (*SRGAP3-A*, *AC104035.1*, *CLIP1*, *LINC0095* and AC005153.1) upregulated and 2 (*ZNF217* and *SUZ12)* downregulated common to the IU and HU comparison while fourteen (13 (*GLOD5*, *AC005703.6*, *SP110*, *PRMT7*, *CCDC141*, *CCDC86*, *AL049647.1*, *CUEDC1*, *AC025278.1*, *FAM170B-AS1*, *KIF1B*, *AL132642.1*) upregulated and 1 (*MALAT1*) downregulated) DEGs were observed in HU and HL comparison and none in IU and HL comparisons.

## Expression of genes associated with stunting and BMI

The mean height for age of the children in the study was in the bottom 3 percent for all children worldwide and 48% of children met the WHO definition of being stunted (height for age $< -2$ SD of global mean). To identify the association of stunting and BMI with gene expression, we conducted a linear regression analysis using height for age Z-scores (HAZ) and body

**Table 2. Summary of number of significant upregulated and downregulated DEGs in the different comparisons.**

| Pair (total observations) | Details | DEGs | Up regulated | Down regulated |
|---|---|---|---|---|
| *All infected vs uninfected (n = 64)* | Infected: 44 Uninfected: 20 | 14 | 9 (64%) | 5 (36%) |
| *High vs uninfected (n = 54)* | High: 34 Uninfected: 20 | 56 | 43 (77%) | 13 (23%) |
| *High vs low infected (n = 44)* | High: 34 Low: 10 | 33 | 30 (91%) | 3 (9%) |
| *Low vs uninfected (n = 30)* | Low: 10 Uninfected: 20 | - | - | - |

**Table 3. List of the significant DEGs between *S. mansoni* infected and uninfected children.**

| GeneName | log2 Fold Change | P value | Adjusted P value | GeneType | State |
|---|---|---|---|---|---|
| CCDC168 | 2.275 | 4.66E-05 | 0.032 | protein_coding | Upregulated |
| AP003498.2 | 1.883 | 1.67E-05 | 0.022 | lncRNA | |
| AC115485.1 | 1.740 | 1.05E-05 | 0.020 | lncRNA | |
| VN1R110P | 1.464 | 8.65E-05 | 0.040 | processed_pseudogene | |
| SRGAP3-AS2 | 1.298 | 3.09E-05 | 0.027 | lncRNA | |
| AC104035.1 | 1.260 | 5.55E-05 | 0.035 | lncRNA | |
| CLIP1 | 1.220 | 1.20E-05 | 0.020 | protein_coding | |
| LINC00945 | 1.070 | 8.81E-05 | 0.040 | lncRNA | |
| AC005153.1 | 1.024 | 8.67E-05 | 0.040 | lncRNA | |
| ZNF217 | -0.874 | 0.00010464 | 0.040 | protein_coding | Downregulated |
| RN7SKP203 | -1.198 | 4.13E-06 | 0.020 | misc_RNA | |
| AC234775.2 | -1.415 | 0.00011825 | 0.044 | processed_pseudogene | |
| SUZ12 | -1.586 | 3.03E-05 | 0.027 | protein_coding | |
| TRBC2 | -1.637 | 8.82E-06 | 0.020 | TRCgene | |

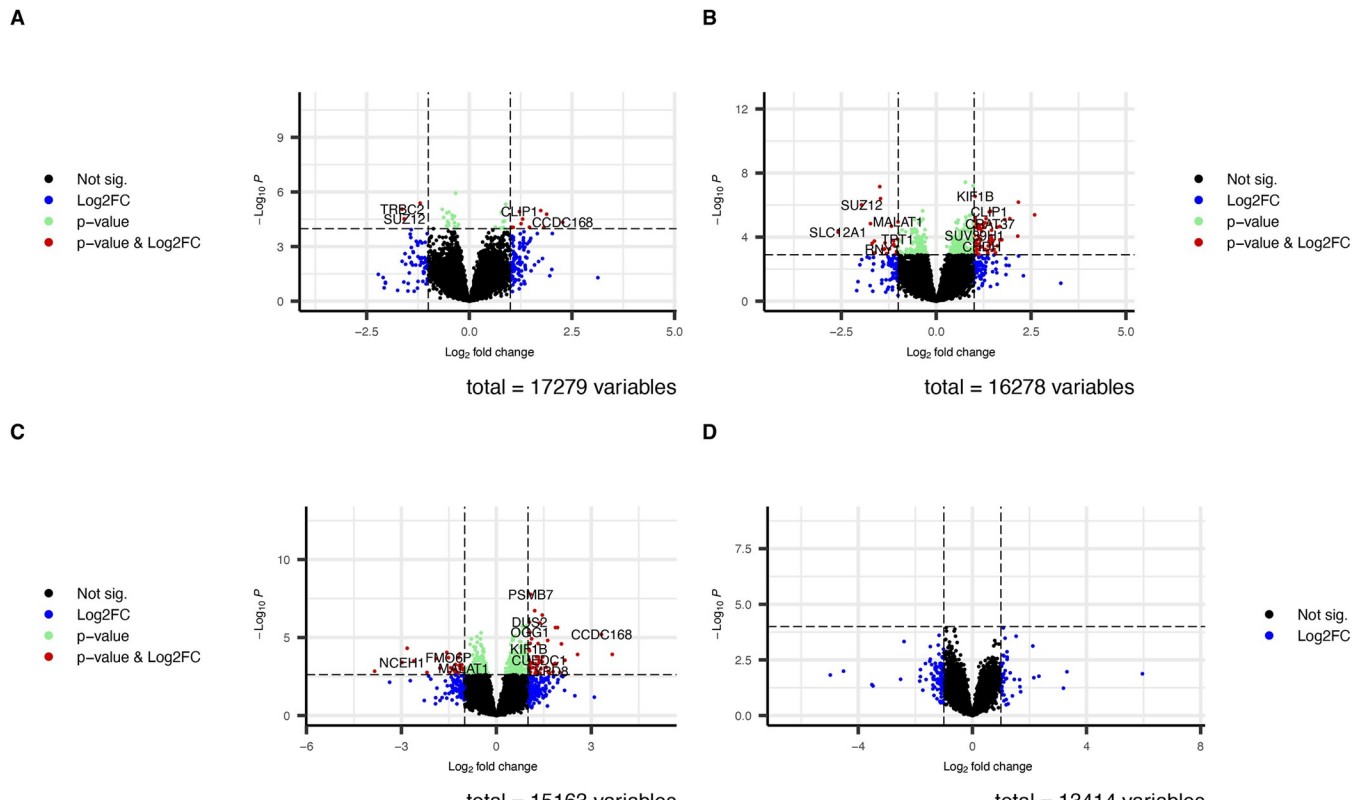

**Fig 3. Volcano plots showing differentially expressed genes between *S. mansoni* infected and uninfected individuals. A** Fourteen (14) significant DEGs (9 upregulated and 5 down regulated) were identified among the *S. mansoni* infected children compared with the uninfected. **B** Fifty-six (56) significant DEGs (43 upregulated and 13 downregulated) were identified among children with high *S. mansoni* infection intensity compared to the uninfected. **C** Thirty-three (33) significant DEGs (30 upregulated and 3 downregulated) were identified among children with high *S. mansoni* infection intensity compared to those with low infection intensity. **D** No significant DEGs were identified among children with low *S. mansoni* infection intensity compared to the uninfected.

**Table 4. Lists the significant DEGs between children with high *S. mansoni* infection intensity compared to uninfected.** A total of 56 (43 upregulated and 13 downregulated) DEGs between children with high *S. mansoni* infection intensity and the uninfected were observed.

| Comparison | GeneName | log2 Fold Change | P value | Adjusted P value | GeneType | GeneDescription |
|---|---|---|---|---|---|---|
| High vs unifected | CCDC168 | 2.593 | 4.09E-06 | 0.004 | protein_coding | coiled-coil domain containing 168 |
| | ZNF385B | 1.701 | 0.00014699 | 0.015 | protein_coding | zinc finger protein 385B |
| | GLOD5 | 1.601 | 2.41E-05 | 0.007 | protein_coding | glyoxalase domain containing 5 |
| | SSPN | 1.499 | 0.000167 | 0.016 | protein_coding | sarcospan |
| | AP000484.1 | 1.484 | 0.00018252 | 0.017 | processed_pseudogene | zinc finger pseudogene |
| | BLM | 1.459 | 3.70E-05 | 0.008 | protein_coding | BLM RecQ like helicase |
| | AC127521.1 | 1.457 | 0.00026549 | 0.021 | lncRNA | novel transcript, antisense to SPNS3 |
| | AC008505.1 | 1.454 | 0.00011933 | 0.014 | lncRNA | novel transcript |
| | SRGAP3-AS2 | 1.433 | 2.45E-06 | 0.004 | lncRNA | SRGAP3 antisense RNA 2 |
| | AC104035.1 | 1.421 | 1.48E-05 | 0.006 | lncRNA | novel transcript |
| | CLIP1 | 1.397 | 2.68E-06 | 0.004 | protein_coding | CAP-Gly domain containing linker protein 1 |
| | AC092745.1 | 1.373 | 0.00035827 | 0.025 | lncRNA | novel transcript |
| | AC005703.6 | 1.316 | 6.95E-06 | 0.005 | TEC | novel transcript |
| | AL049548.1 | 1.287 | 1.27E-05 | 0.006 | lncRNA | novel transcript |
| | SP110 | 1.262 | 2.53E-05 | 0.007 | protein_coding | SP110 nuclear body protein |
| | LINC00945 | 1.238 | 1.59E-05 | 0.006 | lncRNA | long intergenic non-protein coding RNA 945 |
| | CHDH | 1.202 | 0.00039212 | 0.026 | protein_coding | choline dehydrogenase |
| | PRMT7 | 1.200 | 4.40E-05 | 0.008 | protein_coding | protein arginine methyltransferase 7 |
| | SRGAP3-AS3 | 1.174 | 3.02E-05 | 0.007 | lncRNA | SRGAP3 antisense RNA 3 |
| | EPN2-AS1 | 1.169 | 0.00015706 | 0.016 | lncRNA | EPN2 antisense RNA 1 |
| | AL031717.1 | 1.166 | 0.00012754 | 0.014 | lncRNA | novel transcript, antisense to MAPK8IP3 |
| | CCDC141 | 1.146 | 1.89E-05 | 0.006 | protein_coding | coiled-coil domain containing 141 |
| | OLIG1 | 1.131 | 2.32E-05 | 0.007 | protein_coding | oligodendrocyte transcription factor 1 |
| | AC037198.2 | 1.120 | 0.00116746 | 0.048 | lncRNA | novel transcript, antisense to THBS1 |
| | CCDC86 | 1.119 | 8.39E-05 | 0.012 | protein_coding | coiled-coil domain containing 86 |
| | OLIG2 | 1.108 | 3.06E-05 | 0.007 | protein_coding | oligodendrocyte transcription factor 2 |
| | AL049647.1 | 1.104 | 1.64E-05 | 0.006 | lncRNA | novel transcript |
| | CUEDC1 | 1.090 | 0.00025955 | 0.021 | protein_coding | CUE domain containing 1 |
| | ACSM1 | 1.083 | 1.74E-05 | 0.006 | protein_coding | acyl-CoA synthetase medium chain family member 1 |
| | AC091117.2 | 1.078 | 0.0012653 | 0.049 | lncRNA | novel transcript, antisense to SORD |
| | AC005153.1 | 1.073 | 0.00014873 | 0.015 | lncRNA | novel transcript, antisense to GRB10 |
| | AC025278.1 | 1.069 | 0.00030718 | 0.023 | lncRNA | novel transcript, antisense to EMR4P |
| | AC007327.2 | 1.052 | 0.00048388 | 0.030 | lncRNA | novel transcript |
| | AC124283.4 | 1.045 | 0.00070679 | 0.037 | processed_pseudogene | ADP-ribosylation factor-like 2 binding protein (ARL2BP) pseudogene |
| | PGD | 1.033 | 0.00056168 | 0.032 | protein_coding | phosphogluconate dehydrogenase |
| | AC090907.1 | 1.033 | 1.04E-05 | 0.006 | lncRNA | novel transcript, antisense to LRRK1 |
| | FAM170B-AS1 | 1.026 | 6.37E-06 | 0.005 | lncRNA | FAM170B antisense RNA 1 |
| | KIF1B | 1.025 | 2.85E-07 | 0.001 | protein_coding | kinesin family member 1B |
| | AC135048.4 | 1.024 | 0.00020387 | 0.018 | TEC | novel transcript |
| | AL136090.1 | 1.020 | 0.00082991 | 0.040 | lncRNA | novel transcript |
| | SETP11 | 1.019 | 2.53E-05 | 0.007 | processed_pseudogene | SET pseudogene 11 |
| | SUV39H1 | 1.009 | 7.93E-05 | 0.012 | protein_coding | suppressor of variegation 3–9 homolog 1 |
| | AL132642.1 | 1.002 | 1.42E-05 | 0.006 | lncRNA | novel transcript, antisense to ASB2 |
| | ITGA4 | -0.803 | 4.69E-05 | 0.008 | protein_coding | integrin subunit alpha 4 |
| | LINC01712 | -0.805 | 0.00117351 | 0.048 | lncRNA | long intergenic non-protein coding RNA 1712 |
| | OGT | -0.899 | 3.58E-05 | 0.008 | protein_coding | O-linked N-acetylglucosamine (GlcNAc) transferase |
| | A1CF | -0.899 | 0.00099447 | 0.044 | protein_coding | APOBEC1 complementation factor |
| | AKR1A1 | -0.945 | 0.00017262 | 0.016 | protein_coding | aldo-keto reductase family 1 member A1 |
| | MS4A6A | -0.957 | 0.00075701 | 0.038 | protein_coding | membrane spanning 4-domains A6A |
| | ZNF217 | -0.986 | 5.94E-05 | 0.010 | protein_coding | zinc finger protein 217 |
| | MALAT1 | -1.005 | 1.12E-05 | 0.006 | lncRNA | metastasis associated lung adenocarcinoma transcript 1 |
| | TPT1 | -1.018 | 0.00013564 | 0.015 | protein_coding | tumor protein, translationally controlled 1 |
| | MED7 | -1.109 | 0.00034823 | 0.025 | protein_coding | mediator complex subunit 7 |
| | RNY1 | -1.413 | 0.00057363 | 0.032 | misc_RNA | RNA, Ro60-associated Y1 |
| | SUZ12 | -1.968 | 9.71E-07 | 0.003 | protein_coding | SUZ12 polycomb repressive complex 2 subunit |
| | SLC12A1 | -2.587 | 4.89E-05 | 0.008 | protein_coding | solute carrier family 12 member 1 |

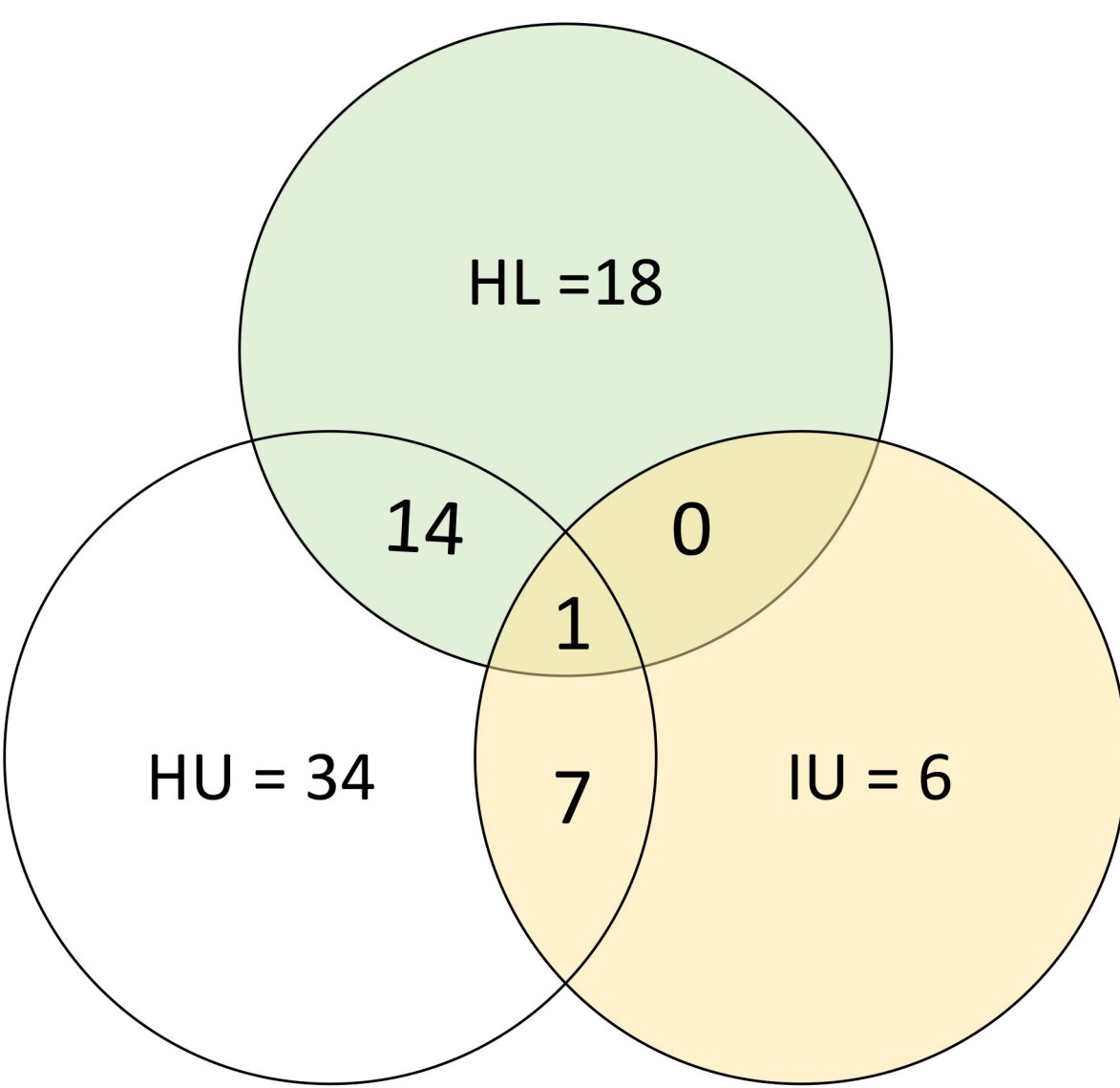

**Fig 4. Counts of DEGs among the different infection intensity comparisons.**

mass index (BMI) scores respectively. We identified significant differential expression of the neutral cholesterol ester hydrolase 1 (*NCEH1*) in stunting which was up regulated with log fold change of 1.28 and p-adj value <0.05. Additionally, we identified four genes with expression that was significantly associated with increase in BMI of which three (*MUC5B*, *DMD* and *REXO1L1P*) were upregulated while one (*SERPINA10*) was downregulated (**Fig 5 and S5 Table**).

## Cell type analysis

To identify the changes in cell type expression with *S. mansoni* infection, the proportions of different cell types in each sample were estimated using Bisque [23]. Bisque analysis uses reference single cell sequence datasets to estimate proportions of different cell types in bulk RNA-seq data. We further used a 2-sided T-test to identify cell types with significant differences in

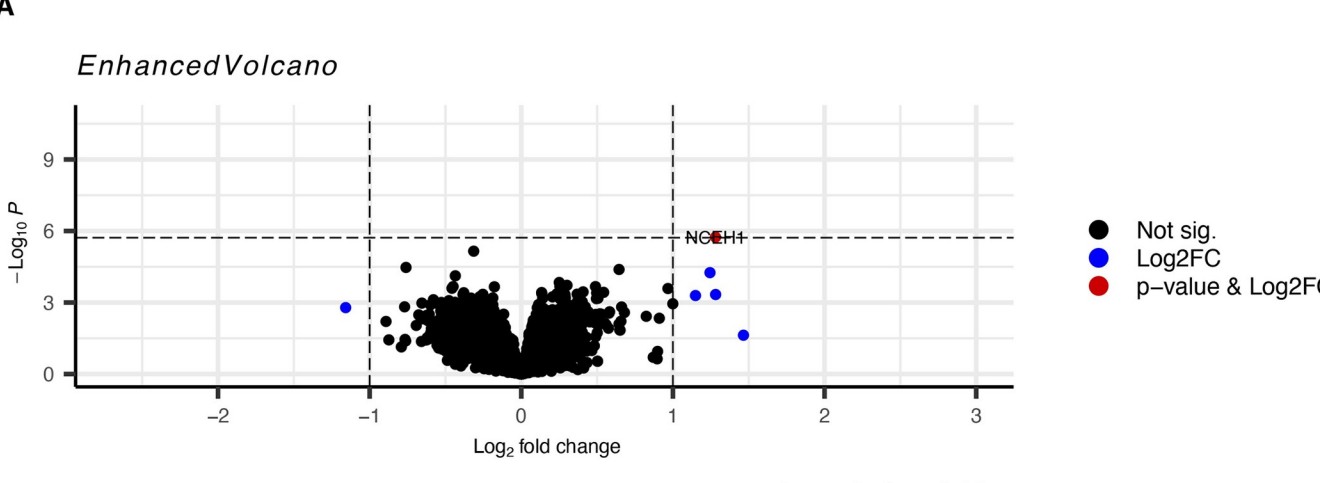

**Fig 5.** **A** Differential expression of genes with stunting. One gene (*NCEH1*) was significantly upregulated in stunting. **B** Differential expression of genes with BMI. Two genes (*MUC5B* and *DMD* were upregulated whereas one gene (*SERPINA10)* was downregulated by increased BMI.

the proportions of each cell type in the blood of infected and uninfected children (**S2 Fig**). Of the 32 cell types evaluated, only Multipotent Progenitors (MPPs) had a significant difference in abundance and were significantly downregulated with a p-value of 0.012 in *S. mansoni* infected individuals compared to the uninfected individuals (**S6 Table**).

## Discussion

Previous studies have shown that mammalian host response to *Schistosoma* infection varies, with some individuals being more susceptible to infection than others. We previously showed high prevalence of *S. mansoni* infection among children living along Lake Albert in Uganda and the same children also had high levels of wasting and stunting although this was not correlated with *Schistosoma* infection [4]. In this study, we present for the first time, data on gene expression in peripheral blood of *S. mansoni* infected children. Gene expression was compared between all infected vs uninfected (IU), highly infected vs uninfected (HU), highly infected vs

low infected (HL), and lightly infected vs uninfected (LU). Similar to findings by Dupnik and colleagues, we observed sufficient difference in gene expression between males and females for the two sexes to cluster separately in the principal components analysis (**Fig 2**) [15]. Like other chronic infections [25,26], we observed only 63 DEGs in the *S. mansoni* infected. Additionally, our data showed gene expression differences when children with high *S. mansoni* infection intensity were compared with low infection intensities and uninfected whereas there was no significant differential gene expression between children with low infection compared to the uninfected. This suggests that gene expression in peripheral blood is scarcely perturbed in individuals with low *S. mansoni* infection. We further observed some gene expression similarities in the different infection intensity comparisons. The *CCDC168* gene was observed in all the comparisons (IU, HU and HL) indicating upregulation of this gene in highly infected individuals compared to the uninfected or those with low infection intensity. There is very limited knowledge about its role [27], but it has been shown to be mutated in several cancers [28]. In the infected and uninfected (IU and HU) comparisons *CLIP1* was upregulated. This gene has been found to mediate microtubule capture through interaction with RHO GTPases [23]. Microtubule capture may be linked to wound healing through stimulation of cell migration and fibroblasts [29,30] and is involved in T Cell regulation [30–32]. This finding is similar to previous murine findings that demonstrated the upregulation of fibrosis linked genes in the late stages of *S. japonicum* infection [11,12,33]. The role of other genes identified with the comparisons is not well known highlighting a need for further investigation of these genes. We searched through Pubmed abstracts and titles using each of the 63 unique differentially expressed gene names and the terms "schistosomiasis" or "schistosoma" and fibrosis.

## Association of DEGs with schistosomiasis or fibrosis

Whilst schistosome infections cause lethargy and other non-specific symptoms, death is mainly caused by the fibrosis accumulating around eggs lodged in the tissues, particularly in the hepatic portal vein. None of the 63 genes had informative associations with schistosomiasis following the Pubmed search with the terms "schistosomiasis" or "schistosoma". The search for abstracts and titles using the term "fibrosis" with each of the 63 genes in turn identified 27 gene names that appeared in articles that also mentioned fibrosis of which 13 had well documented associations with fibrosis (**S3 Table**). Seven genes were also associated with *TGFB1* (**S4 Table**).

Although our study was of whole blood and schistosomiasis associated fibrosis occurs in extracellular matrix, six of the 13 of the well documented DEGS associated with fibrosis had been previously found to be differentially expressed in comparisons of liver fibroses with healthy tissues [34] suggesting that expression data from whole blood could be informative. For 11 genes it was possible to predict a direction of effect on fibrosis that would be caused by a change in gene expression. The expression changes in five genes (*OGG1*, *OGT*, *ITGA4*, *PRMT7*, *SUZ12*), were predicted to increase fibrosis in participants with high parasitaemia and the remaining 6 genes (*MALAT1*, *TPT1*, *A1CF*, *SSPN*, *SUV39H1*, *ZNF217)* were predicted to reduce fibrosis. Therefore, it is not possible to predict the risk of fibrosis in these participants from their expression profiles. A prospective study to determine whether these genes could be useful biomarkers of morbidity risk may be more informative.

*TGFB1* has been described as the master regulator of fibrosis [35]. Although it was not differentially expressed in our study, seven of the 13 genes associated with fibrosis were also associated with *TGFB1*. *SUZ12* suppresses p27 [36] and p27 promotes *TGFB1* mediated pulmonary fibrosis [37]. Furthermore, THBS1 modulates the TGFB1/Smad2/3 signalling pathways [38], OGG1 promotes *TGFB1* induced cell transformation and activated Smad2/3 by

interacting with Smad7 [39]. *ITGA4* and *TGFB1* have been found to be co-expressed in four cancer studies [40–43]. Additionally, *MALAT1* has been found to modulate *TGFB1* induced epithelial to mesenchymal transition in keratinocytes [44] and *TPT1* negatively regulates the *TGFB1* signalling pathway by preventing *TGFB1* receptor activation [45]. These findings indicate an interplay of fibrosis enhancers and modulators in *S. mansoni* infected children living along the Albert Nile in Uganda.

## Gene expression and stunting and BMI

We identified significant differential expression of one gene, the neutral cholesterol ester hydrolase 1 (*NCEH1*) in stunting as measured by HAZ scores. This gene is involved in the breakdown of cholesterol esters in macrophages which makes them available for export and recycling to the liver [46]. It is possible that in stunted children there is a higher rate of cholesterol recycling to make best use of limited lipid resources. Additionally, four DEGS were associated with BMI of which three (*MUC5B*, *DMD* and *REXO1L1P*) were upregulated whereas one (*SERPINA10*) was downregulated. The increased expression of *MUC5B* with BMI is consistent with previous observations on the role of obesity in lung function [47] since increased expression of *MUC5B* has been reported to mediate chronic obstructive pulmonary disease development through regulation of inflammation and goblet cell differentiation [48].

## Changes in cell type frequency

Our analysis of the relative abundance of different cell types showed significant downregulation of Multipotent Progenitor cells (MPPs) in the children infected with *S. mansoni*. MPPs are thought to regulate HSC proliferation in response to inflammation [49] and play a role in regulation of immune response [50]. *S. mansoni* antigens are known to suppress Th1 and Th17 pathways whilst stimulating Th2, B regs and T regs [51]. The reduction in relative abundance of MPPs may contribute to this process.

## Limitations of the study

There were few differentially expressed genes which limited further analysis of pathways affected by the genes. POC-CCA test doesn't differentiate between species and therefore further testing to rule out *S. haematobium* may have been an added advantage although this was not done. To avoid amplification steps, only samples with 1 ug of RNA were sequenced although this could still introduce bias due to exclusion of samples with lower amounts of RNA. Additionally, statistical analysis did not correct for multiple comparisons and an additional laboratory method (e.g. qPCR or other method) to validate these findings would be useful.

## Conclusion

Our study shows evidence of differential expression of genes in *S. mansoni* infection in children living in endemic areas. The low number of differentially expressed genes may be due to *S. mansoni* having to avoid stimulating a strong immune response in order to survive for years in the host. As such the parasite is known to suppress inflammatory responses leading to a relatively weak effect of the parasite on the host transcriptome [51]. Furthermore, many of the children in this study suffered from severe stunting and most were underweight. Malnutrition is associated with increased risk of death from infections and may impair the immune response [52]. Therefore, malnutrition may have reduced the response to *S. mansoni* infection and the number of differentially expressed genes. Importantly we have also identified genes

that may be involved in the development of fibrosis which is the principal pathology associated with schistosomiasis. Follow up studies will be required to determine if expression of these genes correlates with the development of hepatic fibrosis in these children. We also show that there is no significant difference in gene expression between individuals with low levels of infection and those who are uninfected; further studies are required to elucidate this finding.

## Supporting information

**S1 Fig. Principal component analysis of all sequenced samples with CAA results.**
(TIF)

**S2 Fig. Difference in abundance of cell types in *S. mansoni* infected compared to unin-fected individuals.**
(TIF)

**S1 Table. Participant details for the RNAseq analysis.**
(PDF)

**S2 Table. Significant DEGs between children with high *S. mansoni* infection intensity com-pared to those with low infection intensity.**
(PDF)

**S3 Table. DEGs that may be associated with fibrosis.**
(PDF)

**S4 Table. DEGs associated with TGFB1.**
(PDF)

**S5 Table. Expressed genes in stunting and BMI.**
(PDF)

**S6 Table. Cell types that differ in relative abundance between *S. mansoni* infected and uninfected children.**
(PDF)

**S1 Data Analysis Script. RNA Sequence data analysis using DESeq2 pipeline.**
(PDF)

## Acknowledgments

We do acknowledge and thank all the children and the parents/guardians that participated in this study. We do appreciate the efforts by the Village health team members and the local council administrators of the villages of Panyigoro, Kivuje, Nyakagei, Kayonga, Dei, Pamitu and Alwi. Membership of the TrypanoGEN+ Research group of the H3Africa Consortium: Annette MacLeod, Bruno Bucheton, Gustave Simo, Dieudonne N. Mumba, Mathurin Koffi, Ozlem T. Bishop, Pius V. Alibu, Janelisa Musaya, Christiane Hertz-Fowler.

## Author Contributions

**Conceptualization:** Joyce Namulondo, Enock Matovu, Julius Mulindwa.

**Data curation:** Joyce Namulondo, Magambo Phillip Kimuda.

**Formal analysis:** Joyce Namulondo, Oscar Asanya Nyangiri, Peter Nambala, Harry Noyes, Julius Mulindwa.

**Funding acquisition:** Enock Matovu.

**Investigation:** Joyce Namulondo, Harry Noyes, Julius Mulindwa.

**Methodology:** Joyce Namulondo, Julius Mulindwa.

**Resources:** Enock Matovu.

**Supervision:** Harry Noyes, Enock Matovu, Julius Mulindwa.

**Writing – original draft:** Joyce Namulondo.

**Writing – review & editing:** Oscar Asanya Nyangiri, Magambo Phillip Kimuda, Peter Nambala, Jacent Nassuuna, Moses Egesa, Barbara Nerima, Savino Biryomumaisho, Claire Mack Mugasa, Immaculate Nabukenya, Drago Kato, Alison Elliott, Harry Noyes, Robert Tweyongyere, Enock Matovu, Julius Mulindwa.

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
