## [Decision Letter · Decision Letter 0]

22 Jul 2023

Dear Dr Mulindwa,

Thank you very much for submitting your manuscript "Transcriptome analysis of peripheral blood of Schistosoma mansoni infected children from the Albert Nile region in Uganda reveals genes implicated in fibrosis pathology." for consideration at PLOS Neglected Tropical Diseases. As with all papers reviewed by the journal, your manuscript was reviewed by members of the editorial board and by several independent reviewers. In light of the reviews (below this email), we would like to invite the resubmission of a significantly-revised version that takes into account the reviewers' comments. 

This is an interesting manuscript. Please consider the reviewers' thorough comments included, particularly the question about whether the pathways analysis may be revised or removed. It would also be helpful if the authors can provide further information to clarify the study design and the decisions to include or exclude some samples.

We cannot make any decision about publication until we have seen the revised manuscript and your response to the reviewers' comments. Your revised manuscript is also likely to be sent to reviewers for further evaluation.

Sincerely,

Jennifer A. Downs, M.D., Ph.D.

Academic Editor

Eva Clark

Section Editor

This is an interesting manuscript. Please consider the reviewers' thorough comments included, particularly the question about whether the pathways analysis may be revised or removed. It would also be helpful if the authors can provide further information to clarify the study design and the decisions to include or exclude some samples.

Reviewer's Responses to Questions

**Key Review Criteria Required for Acceptance?**

**Methods**

-Are the objectives of the study clearly articulated with a clear testable hypothesis stated?

-Is the study design appropriate to address the stated objectives?

-Is the population clearly described and appropriate for the hypothesis being tested?

-Is the sample size sufficient to ensure adequate power to address the hypothesis being tested?

-Were correct statistical analysis used to support conclusions?

-Are there concerns about ethical or regulatory requirements being met?

Reviewer #1: The authors have provided detailed description of the experiments' methodology.

Reviewer #2: 1. Major: I would like more information about the study. Was screening done as part of a public health effort and then a subset were asked to participate in a research study? When did the research component start – with collection of the screening urine or the blood sample? This could be made clearer in the text and in Figure 1.Were there only 10-15 year olds in the screening? Were they only screened for schistosomiasis or for other parasites? If other parasites, these would be important to include as confounders. The ethics statement includes stool examination for eggs but I didn’t see egg data included for your study participants – were children with positive POC-CCA also treated with praziquantel? If the TrypanoGEN+ study protocol has been published, please reference it as this could clear up a lot of these questions. In the results, there were 727 in “further studies” and 152 in the “gene expression study based on POC-CCA scores.” What was it about the POC-CCA that led those 152 participants to be recruited for this study?

2. Major: CAA can be positive for Schistosoma haematobium and/or S. mansoni. If the Albert-Nile area is endemic only for S. mansoni, I would include that information. If it is endemic for both, did you confirm that eggs were mansoni rather than haematobium? Although it is rare, S. mansoni can cause urogenital schistosomiasis. Did the participants have urine and stool microscopy to assess for eggs?

3. Major: Blood collected in a PAXgene tube usually has RNA extracted using the Paxgene kit, but the authors used Trizol. I am unable to find a published study validating Trizol for RNA extraction from Paxgene tubes, and the reference given (#11) does not specify the RNA extraction method for blood collected in Paxgene tubes. Can the authors elaborate on the rationale for this approach, ideally with a reference validating this method?

4. Minor: Since your audience is not all Schisto experts, would describe what CAA and CCA are and why you did both and whether or not people’s CAA and CCA results needed to be concordant to be included in the study.

5. Minor: Give rationale for assigning someone as low vs. high CAAemia – how did you decide on the cutoff?

6. Minor: Include the details of the POC-CCA used (manufacturer, etc).

7. Minor: Spell out what the UCP-LF of UCP-LF CAA is and include its details (manufacturer, etc). 

8. Typo: line 123 “CAA concentrations > 30 pg/mL” should be “< 30 pg/mL”

9. Typo: line 128 clarify if it should be “quantity of 1 ug” or a “concentration of 1 ug” per what volume. 

10. Typo: line 135 For clarity, would specify 2x150 bp and 30 million

Reviewer #3: -Are the objectives of the study clearly articulated with a clear testable hypothesis stated? Yes

-Is the study design appropriate to address the stated objectives? Yes

-Is the population clearly described and appropriate for the hypothesis being tested? Yes

-Is the sample size sufficient to ensure adequate power to address the hypothesis being tested? This study is a hypothesis generating type of study. So no problem.

-Were correct statistical analysis used to support conclusions? Yes

-Are there concerns about ethical or regulatory requirements being met? No

**Results**

-Does the analysis presented match the analysis plan?

-Are the results clearly and completely presented?

-Are the figures (Tables, Images) of sufficient quality for clarity?

Reviewer #1: The results are clearly.

Reviewer #2: 1. Major: 80 samples were “selected for sequencing to represent the extremes of infection intensity” which I assumed meant their CAA-determined infection intensity, yet “11 lacked CAA results and were excluded.” Could the authors elaborate on this? Additionally, only 52% of RNA passed QC, which seems very low to me, and I am assuming is because of the 1 ug cutoff. Why was the cutoff so high? The amount of input RNA for RNA-Seq is usually much lower than 1 ug. My concern would be that participants with lower WBC counts would be excluded since they wouldn’t generate enough RNA to meet that cutoff which could skew your results.

2. Major: While it is inconvenient when there are people who do not cluster in a PCA, I’m not sure that excluding 5 people is the right approach, nor do the reasons given seem appropriate. We don’t usually exclude people from analysis because of their ethnicity. Poor RNA quality should have been detected with Qubit and Agilent bioanalyzer QC before sequencing. Which schisto groups were these 5 people in?

3. Major: There are so few differentially expressed genes, and so few that are included in the Reactome database, that I don’t think that pathways analysis is an appropriate approach. There is only 1 gene included in each pathway of Tables 4 and S4 and 1 or 2 genes for each pathway in Table S8. Although the p-value numbers indicate statistical significance, I question if 1 differentially expressed gene is enough to invoke pathologic importance for that pathway. It may be that there really aren’t that many differences in gene expression between the groups and that alone is the conclusion – adding pathways analysis seems like trying to create a difference where there might not be one.

4. Major: The BMI and stunting subanalyses are only done in the schistosomiasis group – why is this? In discussion, would need to clarify that the associations are for BMI and stunting in people with schistosomiasis at the time of the study. Were these studies done for a subset of the differentially expressed genes or an analysis of all of the RNASeq data?

5. Major: The inclusion of the PubMed search results as part of methods and results for specific gene products, while thorough, would seem to be more appropriate for integration into the discussion where there are relevant citations, for example the authors do this in lines 288-290 with reference 28.

6. Minor: How did the authors compensate for the multiple comparisons – 4 between-group comparisons (neg vs pos, high vs low, high vs neg, low vs neg) – in deciding on a cutoff for statistical significance?

7. Minor: Was the regression analysis for differential gene expression only done for the subset of genes already found to be differentially expressed in the S. mansoni group?

Reviewer #3: -Does the analysis presented match the analysis plan? Yes

-Are the results clearly and completely presented? No

-Are the figures (Tables, Images) of sufficient quality for clarity? No

**Conclusions**

-Are the conclusions supported by the data presented?

-Are the limitations of analysis clearly described?

-Do the authors discuss how these data can be helpful to advance our understanding of the topic under study?

-Is public health relevance addressed?

Reviewer #1: This article provides sufficient information to understand the genetic changes in human children infected with Schistosoma mansoni. The results have potential implications for the diagnosis and treatment of schistosomiasis.

Reviewer #2: Because of the aforementioned questions/concerns about the data analysis approach, it is difficult to assess the conclusions. The limitations noted by this reviewer were not considered in the conclusion as written. While the findings of association of some of the differentially expressed genes in the schistosomiasis group with fibrosis is interesting, there were so few differentially expressed genes (and only 4 protein-coding) that the conclusions may have been overstated.

Reviewer #3: -Are the conclusions supported by the data presented? Yes

-Are the limitations of analysis clearly described? Yes

-Do the authors discuss how these data can be helpful to advance our understanding of the topic under study? Not much, should have mentioned about the benefit for future diagnosis and treatment.

-Is public health relevance addressed? No

**Editorial and Data Presentation Modifications?**

Reviewer #1: Accept

Reviewer #2: 1. Major -- add relevant available demographic data to Table 1 – age, coinfections, BMI, stunting, what subcounty site a person was from, average CAA, % with S. mansoni eggs seen in stool, etc. Any significant differences in these variables should be addressed.

2. Major – consider if the fold changes, adjusted p-values, and FDR should/need to be taken to the millionths place in Tables 3, 5, S2, S3, S4. 

3. Major – Figures 3 and 4 are illegible (cannot assess contents given small text size and low resolution)

4. Please specify where the RNA-Seq data are deposited.

5. Consider providing the code used for analysis as supplemental material or a link to a github with that information.

Reviewer #3: Minor modification is enough.

**Summary and General Comments**

Reviewer #1: In this manuscript, the authors investigated the RNA sequencing of 44 Schistosoma mansoni-infected children and 20 uninfected children, with the reads aligned to the GRCh38 human genome. The study is well-conducted, and its scientific soundness is reasonable. However, before publication, some points should be addressed, which are discussed below:

1. It is recommended to reconfirm the RNA expression of significant DEGs (Differentially Expressed Genes) using qPCR, especially for CLTP1, SUZ12, and TRBC2.

2. Apart from CLTP1, SUZ12, and TRBC2 genes, more detailed explanations should be provided for other significant DEGs identified in the study.

Reviewer #2: In this manuscript, Namulondo and colleagues use transcriptomics to compare peripheral blood gene expression in young adolescents with or without schistosomiasis and with low vs. high schistosomal burden. I appreciate the important work to describe the human immune status during schistosomal infection using gene expression studies. The main finding may be that there aren’t many alterations in gene expression in the peripheral blood – in itself, an interesting finding! Namulondo et al. do mention that it may be that the worms don’t induce an immune response, but then go on to do the pathways and other analyses to try to find patterns in 14 differentially expressed genes, of which only 4 were protein encoding, with statistically significant pathways only including one of the differentially expressed genes. I would recommend focusing on the similarities between schisto/no schisto rather than the differences.

Reviewer #3: This paper is designed to characterize different types of Schistosoma mansoni infected persons in highly endemic area in Uganda by RNAseq analysis using peripheral blood. Density of parasite load was evaluated by circulating antigen (CCA and CAA) levels. High, Low, and No infection groups with stunting and BMI, Blood cell types were well defined. their conclusion that high levels of active infection influenced the expression levels of limited number of genes that were mainly related to fibrotic and TGF-B1 activation. Although the results are simply descriptive, this transcriptome information will be beneficial for future development of relevant research. 

There are several comments to be improved.

1. Fig.3 and Fig 4 needs clearer picture with high resolution. And more important is to make more easy-to-understand legends that explain the details of the methodology.

2. Cell types analysis needs more explanation. How the authors identified MPP showed a significant change between the groups?

3. There was no description of antibody levels of the individuals. If available, please describe it.

4. Generally, fibrosis in the liver is frequently observed in the endemic area? Even younger generation?

PLOS authors have the option to publish the peer review history of their article (what does this mean?). If published, this will include your full peer review and any attached files.

Reviewer #1: No

Reviewer #2: No

Reviewer #3: No
---

## [Editor Report · Decision Letter 1]

10 Oct 2023

Dear Dr Mulindwa,

Thank you very much for submitting your manuscript "Transcriptome analysis of peripheral blood of Schistosoma mansoni infected children from the Albert Nile region in Uganda reveals genes implicated in fibrosis pathology." for consideration at PLOS Neglected Tropical Diseases. As with all papers reviewed by the journal, your manuscript was reviewed by members of the editorial board and by several independent reviewers. The reviewers appreciated the attention to an important topic. Based on the reviews, we are likely to accept this manuscript for publication, providing that you modify the manuscript according to the review recommendations. 

We thank the authors for their responsiveness to reviews and the major edits that they have made, which have substantially improved the manuscript. There are still some unresolved issues that need correction:

The results that no other parasites were found in the stool of the children selected for RNA-Seq is not clearly presented and should be mentioned in the text.

The ethics statement states that children who had eggs in stool were treated with praziquantel, but there is still no comment about those who were POC-CCA positive being treated.

Reference 16 reports CAA cut-offs that are different than what was used in this manuscript. Please reconcile this. Also, in lines 148-150 there is not a clear classification for what would happen if someone’s CAA result was exactly 1000 pg/mL.

The EGA number does not appear to be included in the text anywhere.

The discussion does not include a paragraph about limitations of this study, which were pointed out by reviewers. Please add this and be sure to include:

-POC-CCA doesn’t differentiate between species; urine was not filtered. 

-Please mention the requirement that samples had to have 1 ug of RNA, which was done to avoid amplification steps but could still introduce bias because of excluding those with lower amounts of RNA.

-Statistical analysis did not correct for multiple comparisons and an additional laboratory method (e.g. qPCR or other method) to validate these findings would be useful.

Sincerely,

Jennifer A. Downs, M.D., Ph.D.

Academic Editor

Eva Clark

Section Editor

We thank the authors for their responsiveness to reviews and the major edits that they have made, which have substantially improved the manuscript. There are still some unresolved issues that need correction:

The results that no other parasites were found in the stool of the children selected for RNA-Seq is not clearly presented and should be mentioned in the text.

The ethics statement states that children who had eggs in stool were treated with praziquantel, but there is still no comment about those who were POC-CCA positive being treated.

Reference 16 reports CAA cut-offs that are different than what was used in this manuscript. Please reconcile this. Also, in lines 148-150 there is not a clear classification for what would happen if someone’s CAA result was exactly 1000 pg/mL.

The EGA number does not appear to be included in the text anywhere.

The discussion does not include a paragraph about limitations of this study, which were pointed out by reviewers. Please add this and be sure to include:

-POC-CCA doesn’t differentiate between species; urine was not filtered. 

-Please mention the requirement that samples had to have 1 ug of RNA, which was done to avoid amplification steps but could still introduce bias because of excluding those with lower amounts of RNA.

-Statistical analysis did not correct for multiple comparisons and an additional laboratory method (e.g. qPCR or other method) to validate these findings would be useful.

Figure Files:

Data Requirements:

Reproducibility:

References

---

## [Editor Report · Decision Letter 2]

3 Nov 2023

Dear Dr Mulindwa,

We are pleased to inform you that your manuscript 'Transcriptome analysis of peripheral blood of Schistosoma mansoni infected children from the Albert Nile region in Uganda reveals genes implicated in fibrosis pathology.' has been provisionally accepted for publication in PLOS Neglected Tropical Diseases.

Best regards,

Jennifer A. Downs, M.D., Ph.D.

Academic Editor

Eva Clark

Section Editor

---

## [Editor Report · Acceptance letter]

9 Nov 2023

Dear Dr Mulindwa,

We are delighted to inform you that your manuscript, "Transcriptome analysis of peripheral blood of Schistosoma mansoni infected children from the Albert Nile region in Uganda reveals genes implicated in fibrosis pathology.," has been formally accepted for publication in PLOS Neglected Tropical Diseases.

Best regards,

Shaden Kamhawi

co-Editor-in-Chief

Paul Brindley

co-Editor-in-Chief
